# The Fate of Oxidative Strand Breaks in Mitochondrial DNA

**DOI:** 10.3390/antiox12051087

**Published:** 2023-05-12

**Authors:** Genevieve Trombly, Afaf Milad Said, Alexei P. Kudin, Viktoriya Peeva, Janine Altmüller, Kerstin Becker, Karl Köhrer, Gábor Zsurka, Wolfram S. Kunz

**Affiliations:** 1Institute of Experimental Epileptology and Cognition Research, Medical Faculty, University of Bonn, 53127 Bonn, Germany; 2Cologne Center for Genomics and Center for Molecular Medicine Cologne (CMMC), Medical Faculty, University of Cologne, 50923 Köln, Germany; 3Institute of Human Genetics, University of Cologne, 50923 Köln, Germany; 4Biological and Medical Research Centre (BMFZ), Genomics & Transcriptomics Laboratory, Heinrich-Heine-University Duesseldorf, 40225 Düsseldorf, Germany; 5Department of Epileptology, University Hospital Bonn, 53127 Bonn, Germany

**Keywords:** mitochondrial DNA, oxidative damage, mtDNA double-strand breaks mtDNA single-strand breaks, mtDNA degradation

## Abstract

Mitochondrial DNA (mtDNA) is particularly vulnerable to somatic mutagenesis. Potential mechanisms include DNA polymerase γ (POLG) errors and the effects of mutagens, such as reactive oxygen species. Here, we studied the effects of transient hydrogen peroxide (H_2_O_2_ pulse) on mtDNA integrity in cultured HEK 293 cells, applying Southern blotting, ultra-deep short-read and long-read sequencing. In wild-type cells, 30 min after the H_2_O_2_ pulse, linear mtDNA fragments appear, representing double-strand breaks (DSB) with ends characterized by short GC stretches. Intact supercoiled mtDNA species reappear within 2–6 h after treatment and are almost completely recovered after 24 h. BrdU incorporation is lower in H_2_O_2_-treated cells compared to non-treated cells, suggesting that fast recovery is not associated with mtDNA replication, but is driven by rapid repair of single-strand breaks (SSBs) and degradation of DSB-generated linear fragments. Genetic inactivation of mtDNA degradation in exonuclease deficient POLG p.D274A mutant cells results in the persistence of linear mtDNA fragments with no impact on the repair of SSBs. In conclusion, our data highlight the interplay between the rapid processes of SSB repair and DSB degradation and the much slower mtDNA re-synthesis after oxidative damage, which has important implications for mtDNA quality control and the potential generation of somatic mtDNA deletions.

## 1. Introduction

The idea of persistent oxidative damage of mtDNA leading to the accumulation of somatic mutations during a lifetime has led to the ‘mitochondrial theory of aging’ [1]. This is based on the fact that mtDNA is localized at the matrix side of the mitochondrial inner membrane—a highly oxidative environment due to the presence of iron required for heme and iron-sulfur protein synthesis [2] as well as the presence of single-electron donors within the mitochondrial respiratory chain required for the formation of reactive oxygen species (ROS) [3]. The ‘mitochondrial theory of aging’ concept has been challenged by later observations that the somatic mitochondrial point mutation profile of aging cells and tissues does not contain abundant G > T/C > A transversions that are considered to be the hallmark of oxidative DNA damage. Rather, abundant G > A/C > T transitions point to mitochondrial DNA polymerase γ (POLG) errors as the potential cause of the somatic mutation profile in aging [4,5]. Recent publications have accordingly paid little attention to the role of oxidative mutagenesis in somatic mtDNA mutation generation [6,7,8]. However, it has been noted that the missing G > T/C > A point mutation profile does not exclude profound oxidative mutagenesis of mtDNA, but can be explained by the nucleotide selectivity of POLG, which allows 7,8-dihydro-8-oxo-20-deoxyguanosine (8oxodG, an abundant product of DNA oxidation) to pair with cytosine [9]. Additionally, the highly reactive ^•^OH radical does not only modify bases but can also generate mtDNA single-strand breaks (SSB) or a combination of both (for a comprehensive overview of products, see [10]). This process eventually leads to double-strand breaks (DSB) which, in turn, might facilitate deletion formation. To avoid this harmful somatic mutagenesis, mitochondria contain (i) a LIG3-dependent base excision repair (BER) pathway that includes repair of single strand breaks [11], and (ii) a MGME1 and POLG exonuclease-dependent degradation pathway of linear mtDNA [12]. mtDNA degradation after oxidative stress has been suggested to circumvent the accumulation of somatic mutations leading to the idea of a ‘disposable mitochondrial genome’ [13,14]. However, the relative impact of repair and degradation pathways for mtDNA quality control still remains to be elucidated.

In the present report, we describe the interplay of repair, degradation and re-synthesis of mtDNA in an immortalized human embryonic kidney (HEK 293) cell model after oxidative damage by transient hydrogen peroxide treatment. We show that efficient degradation of DSB-generated linear DNA is pivotal to mtDNA quality control and ensures, together with fast repair of SSBs, effective clearance of oxidatively damaged mtDNA molecules.

## 2. Materials and Methods

### 2.1. Cell Culture

Immortalized human embryonic kidney (HEK 293) cells were commercially obtained from ATCC, catalogue Nr. CRL-1573. Cells were cultured in DMEM with GlutaMAX^TM^ (Gibco, Paisley, UK) or stable glutamine (PAN Biotech, Aidenbach, Germany) containing high glucose (25 mM) and 1 mM sodium pyruvate. The medium was supplemented with 50 μg/mL uridine (Sigma-Aldrich/Merck, Darmstadt, Germany), 10% heat-inactivated tetracycline-free FBS (PAN Biotech, Aidenbach, Germany), and 100 U/mL penicillin and streptomycin (Gibco, New York, NY, USA). The POLGexo^−/−^ cell line (p.D274A mutant) was obtained by CRISPR/Cas9 genome editing as described in Ref. [12]. The genotype was confirmed by Sanger sequencing (Appendix A).

Transient oxidative stress was induced on cells seeded at 90% confluence by applying H_2_O_2_ (Honeywell, Seelze, Germany) at concentrations of 0.5 or 1 mM. Viability was determined by adding 0.1% erythrosine B (Sigma-Aldrich, St. Louis, MO, USA) to cells suspended in 1× PBS (Gibco, Paisley, UK) and using a Neubauer hemocytometer (Paul Marienfeld, Lauda-Königshofen, Germany).

### 2.2. Measurement of H_2_O_2_ Decay

Time-dependent concentration changes of H_2_O_2_ in cell growth medium in the presence or absence of cells were determined by the fluorometric Amplex red/peroxidase-coupled method [15] using 1 μM Amplex red (Sigma-Aldrich, St. Louis, MO, USA) and 20 U/mL horseradish peroxidase (Sigma-Aldrich, St. Louis, MO, USA) at λ_ex_ = 560 nm and λ_em_ = 590 nm. For the measurement, a phenol red-free cell culture medium was used.

### 2.3. Total DNA Isolation

DNA was isolated from cell pellets with the QIAamp DNA mini kit (Qiagen, Hilden, Germany) according to the manufacturer’s protocol for tissue isolation. DNA concentration was measured using a Qubit™ 4 fluorometer (Invitrogen/Fisher Scientific, Schwerte, Germany). Qubit™ assay tubes and a Qubit™ 1× dsDNA HS Assay Kit (Invitrogen, Eugene, OR, USA) were used according to the manufacturer’s protocol.

### 2.4. Isolation of mtDNA

Mitochondria were isolated from whole cells via differential centrifugation based on the protocol outlined in [16]. Before mitochondria isolation, ice-cold isolation buffer (210 mM mannitol, 70 mM sucrose and 5 mM HEPES-KOH, pH = 7.2) was bubbled with argon gas (Linde, Pullach, Germany) for a minimum of 30 min to remove any oxygen in the solution that would result in additional oxidative mtDNA damage and supplemented with 0.25% BSA (PAN Biotech, Aidenbach, Germany). The mitochondrial pellet obtained from 300 to 900 million cells was incubated with proteinase K (Qiagen, Hilden, Germany, >600 mAU/mL; 15 µL in 900 µL argon-saturated isolation buffer without BSA) at 26 °C for 1 h. Mitochondrial pellets were washed twice in BSA-containing isolation buffer and directly underwent DNA isolation using the QIAamp DNA mini kit (Qiagen, Hilden, Germany).

### 2.5. Southern Blot

A total of 1 µg DNA was digested with *Mlu*I-HF restriction endonuclease (New England Biolabs, Frankfurt am Main, Germany) for cleavage of nuclear DNA. This restriction endonuclease does not have a cutting site on the mitochondrial genome. DNA was separated, along with DIG-labeled DNA Molecular Weight Marker II (Roche, Mannheim, Germany), in a 0.6% agarose gel containing 1.25 µM ethidium bromide at 40 V overnight. The gels were alkaline treated and neutralized before the DNA was blotted to Zeta-Probe membranes (Bio-Rad, Hercules, CA, USA) and immobilized by baking at 80 °C for 30 min. Blots were hybridized overnight at 48 °C with PCR-generated digoxigenin-labeled probes. Probes were synthesized using the PCR DIG Probe Synthesis Kit (Roche, Mannheim, Germany) with primers 5′-TCATCCCTGTAGCATTGTTCG-3′ and 5′-GAAGAACTGATTAATGTTTGGGTCT-3′ for the *MT-ND5* gene in the region 12602–12690 in the mitochondrial genome, or primers 5′-GTTGGTGGAGCGATTTGTCT-3′ and 5′-GGCCTCACTAAACCATCCAA-3′ for nuclear 18S rRNA genes. Chemiluminescent detection with anti-DIG-AP antibody F_ab_ fragment (Roche, Mannheim, Germany) and CSPD (Roche, Mannheim, Germany) was performed, and the signal was recorded on a ChemiDoc Imaging System (Bio-Rad, Hercules, CA, USA) [12].

### 2.6. Assessment of the Number of mtDNA Breaks by qPCR

Quantitative PCR (qPCR) was used to determine the relative amount of damage present in mtDNA by comparing a short mtDNA product to one over a longer region of mtDNA (gene accession number: NM_001126131.2 and NP_001119603.1). Primers 3922F (5′-CTAGGAAGATTGTAGTGGTGAGGGTG-3′) and 4036R (5′-GAACTAGTCTCAGGCTTCAACATCG-3′) were used to amplify a minor arc segment 115 bps in size, while primers 3922F together with 5625R (5′-ACACCGCTGCTAACCCCATAC-3′) amplify 1704 bps of the minor arc. Amplifications were performed in a C1000 Touch™ Thermal Cycler CFX96™ Real-Time System (Bio-Rad, Hercules, CA, USA) using Luna Universal qPCR Master Mix (New England Biolabs, Frankfurt am Main, Germany) under the following conditions: 95 °C for 15 min and 38 cycles of 95 °C for 15 s, 70 °C for 30 s and 68 °C for 4 min. Triplicate reactions were performed for three different concentrations of template DNA. C_t_ values were defined at the inflection points of fitted 4-parameter sigmoidal or Chapman curves. The relative frequency of breaks per molecule was calculated according to the formula ln(2^((C_t[long]_ − C_t[short]_) − ΔC_t[ref]_)) where ΔC_t[ref]_ is the average value of C_t[long]_ − C_t[short]_ in wild-type cells before treatment. Amplification efficiency was 97.5 ± 1.4% for the short fragment and 75.5 ± 2.8% for the long fragment.

### 2.7. BrdU Incorporation

Incorporation of 5-bromo-2′-deoxyuridine (BrdU) into mtDNA can be used to visualize replication and repair kinetics [17]. One hour before applying BrdU to cells, aphidicolin (Merck, Darmstadt, Germany) dissolved in DMSO was added to the cell culture medium at a final concentration of 20 µM in order to allow for the halting of nuclear DNA replication [18]. Water-dissolved BrdU (Roche, Mannheim, Germany) was then added to the cell culture medium to a final concentration of 10 µM. BrdU incorporation was visualized by Southern blotting and chemiluminescent detection using an anti-BrdU primary antibody (mouse IgG, Product no: 11170376001, Merck, Darmstadt, Germany) and a goat anti-mouse HRP conjugate (Bio-Rad, Hercules, CA, USA) as a secondary antibody. The chemiluminescent signal was developed using the Clarity Western ECL Substrate kit (Bio-Rad, Hercules, CA, USA), and the signal was recorded on a ChemiDoc Imaging System (Bio-Rad, Hercules, CA, USA) [19].

### 2.8. Short-Read (Illumina) Sequencing of Linker-Ligated Isolated mtDNA

A total of 1.5 µg purified mtDNA was ligated to a one-side-blunt double-stranded linker as described in [12] and subsequently column purified with QIAamp DNA Mini Kit (Qiagen, Hilden, Germany). Libraries were prepared and size selected by using the Illumina TruSeq Nano DNA Sample Preparation Kit and Agencourt AMPure XP beads. One cycle of PCR followed to complete the library adaptor structure. Libraries were validated with the Agilent 2200 TapeStation and quantified by qPCR. An Illumina NovaSeq 6000 instrument (Illumina, San Diego, CA, USA) generated 150 bp paired-end reads. For each sample, 0.7–1.4 × 10^7^ paired mitochondrial reads were obtained, representing 50–95% of all reads and resulting in 1.2–2.3 × 10^5^ average coverage. Reads were aligned to sample-specific reference mitochondrial sequences and screened for the linker sequence using an in-house Perl script (available upon request). To convert potential overhangs at DSBs to ligatable blunt ends and to increase the efficiency of linker ligation, mtDNA samples were treated with T4 polymerase and polynucleotide kinase using the Quick Blunting™ Kit (New England Biolabs, Frankfurt am Main, Germany) according to the manufacturer’s protocol: 1 µL of enzyme mix was used to treat 3 µg of mtDNA for 30 min at room temperature. To aid in the detection of SSBs, 350 ng mtDNA was treated with 100 U S1 nuclease (Thermo Scientific, Vilnius, Lithuania) for 15 min at 37 °C [20] before proceeding with linker ligation as described above.

### 2.9. PacBio Single-Molecule Long-Read Sequencing of Isolated mtDNA

For PacBio single-molecule long-read sequencing, 1.5 µg of isolated mtDNA was linearized by the single-cutter restriction endonuclease *Eag*I (New England Biolabs, Frankfurt am Main, Germany). Samples were RNase digested and purified on 0.8× AMPure PB beads prior to library preparation with the Express Template 2.0 kit (Pacific Biosciences, Menlo Park, CA, USA)—according to the protocol “Procedure & Checklist—Preparing Multiplexed Microbial Libraries Using SMRTbell Express Template Prep Kit 2.0” (Version 04, November 2019)—without additional DNA shearing, starting with the removal of single-strand overhangs, and using the Barcoded Overhang Adapter Kit 8A or B (Pacific Biosciences) for adapter ligation. After library preparation, six libraries were pooled equimolarly, and the pools’ size selected with diluted AMPure PB beads to remove fragments <3 kb. Library pools were quantified (Qubit) and analyzed for final fragment size distribution (Fragment Analyzer, Agilent, Santa Clara, CA, USA). Sequencing primers and polymerase were successively annealed and bound to the libraries, and each pool was sequenced on one 8M SMRT cell on a Sequel II Instrument (Pacific Biosciences, Menlo Park, CA, USA) with 30 h movie time and 2 h pre-extension. Circular consensus sequence reads were generated and demultiplexed with SMRT Link v9 (Pacific Biosciences, Menlo Park, CA, USA). S1 nuclease treatment was used to detect SSBs as described for Illumina-based sequencing. Per sample, 5–22 × 10^4^ reads were obtained. Long-reads were aligned using an in-house R script based on the pairwiseAlignment function of the Biostrings package (version 2.64.1) with parameters gapOpening = 5, gapExtension = 2 and nucleotide substitution matrix of match = 1, mismatch = −3, baseOnly = true. Only ends with a relative frequency of >3 × 10^−5^ were used for further analysis.

## 3. Results

### 3.1. Transient Hydrogen Peroxide Treatment of HEK 293 Cells Causes Reversible Oxidative Damage of Mitochondrial DNA

In order to model the oxidative damage of mtDNA, we treated HEK 293 cells with 0.5 mM or 1 mM hydrogen peroxide. Due to the presence of the antioxidant pyruvate in the cell growth medium [21], the H_2_O_2_ concentration rapidly decreased even in the absence of cells (Appendix A). When cells were present, intrinsic cellular H_2_O_2_ splitting activities further accelerated the H_2_O_2_ decay and led to complete removal within 10 min. Thus, our treatment is equivalent to a short H_2_O_2_ pulse and did not substantially alter cellular viability (>65% at each time point, Appendix A).

To evaluate the integrity of mtDNA after H_2_O_2_-induced oxidative damage, we performed Southern blotting of agarose electrophoresis separated DNA under non-denaturing conditions (Figure 1). In the presence of ethidium bromide in the agarose gel, intact circular mtDNA migrates as ‘supercoiled’ faster than full-length (16.5 kb) linear molecules, while nicks, gaps and other alterations can lead to the relaxation of the circular DNA and generate ‘open circles’ that move slower than the full-length linear mtDNA (Figure 1A,B, lane 1) [9]. Both concentrations of H_2_O_2_ (0.5 mM and 1 mM) caused an immediate decrease of mtDNA in supercoiled conformation, while the amounts of linear mtDNA and open circle mtDNA increased (Figure 1A,B, lane 2). At 1 mM H_2_O_2_ concentration, highly fragmented mtDNA was detectable as a smear (Figure 1B, lane 2, Figure 1J). At later time points, the amount of supercoiled mtDNA increased and led to a nearly full recovery after 2 h in 0.5 mM H_2_O_2_-treated cells and after 6 h at 1 mM H_2_O_2_ (Figure 1C,E). In parallel, the amount of linear and open-circle species decreased with time (Figure 1A,B, lanes 3–6).

Previously, we identified exonuclease-mediated degradation as an important pathway for the clearance of linear mtDNA [12]. In our original HEK 293 cell model, DSBs were introduced by mitochondria-targeted restriction endonucleases, and we found that mtDNA degradation was blocked by the lack of MGME1 exonuclease or by the presence of exonuclease-deficient DNA polymerase γ. In order to evaluate the relevance of the mtDNA degradation pathway in the condition of oxidative damage, we compared the effects of H_2_O_2_ treatment in wild-type HEK 293 cells with cells harboring the p.D274A mutation in POLG that leads to inactivation of the 3′-5′ exonuclease activity but leaves the polymerase activity intact [12,22]. Similar to wild-type cells, supercoiled mtDNA disappeared after H_2_O_2_ treatment in POLGexo^−/−^ cells (Figure 1A,B lane 8, Figure 1C,E). However, linear mtDNA (both full-length and highly fragmented) persisted in POLGexo^−/−^ cells and the recovery of supercoiled mtDNA was severely diminished (Figure 1A,B lanes 9–12, Figure 1C,E). Notably, we detected highly fragmented mtDNA species in POLGexo^-/-^ cells even before the H_2_O_2_ treatment (Figure 1A,B, lane 7), which is in concert with observations in other studies [22,23].

In order to observe a stronger effect of oxidative damage on mtDNA integrity and because cell survival was not strongly affected, the 1 mM H_2_O_2_ concentration was chosen for further experimentation. We confirmed the transient appearance of mtDNA breaks after the 1 mM H_2_O_2_ pulse by quantitative real-time PCR. In this approach, amplification efficiencies of a short (115 bp) and a long (1.7 kb) amplicon are compared (Figure 2). A larger observed C_t_ difference between both amplicons corresponds to a greater amount of mtDNA molecules carrying at least one break within the region spanned by the long amplicon. In line with the Southern blot data presented in Figure 1, the mtDNA damage peaked 30 min after the hydrogen peroxide pulse and disappeared almost 24 h after treatment in wild-type HEK 293 cells (Figure 2, blue bars). In the POLGexo^−/−^ cell line, the mtDNA damage persisted at much higher levels (Figure 2, grey bars). The qPCR amplification on native DNA detects both DSBs and SSBs, the latter at half efficiency because one strand is intact. We aimed to quantify SSBs in 0 h and 24 h samples by digesting DNA with S1 nuclease prior to amplification. S1 nuclease removes single-stranded regions from double-stranded DNA not only at ends but also within the DNA. By the activity of the enzyme, intact strands are cut opposite to nicks and gaps, thus, converting SSBs to DSBs. We estimated the number of SSBs by subtracting non-S1 nuclease-treated values from S1 nuclease-treated values. In opposite to total strand breaks (DSBs + SSBs), SSBs did not show a significant difference either at 0 h (wild-type, 0.76 ± 0.16; POLGexo^−/−^, 0.53 ± 0.01; *p* = 0.38) or 24 h (wild-type, 0.57 ± 0.18; POLGexo^-/-^, 0.55 ± 0.15; *p* = 0.96) between wild-type and POLGexo^−/−^ cells. This suggests that SSB repair is unlikely to be influenced by the lack of exonuclease activity of mutant POLG. Note that, using the qPCR method applied in this study, it is not possible to compare amounts of SSBs at the 30 min time point because even non-S1 nuclease-treated values are close to saturation in the POLGexo^−/−^ sample.

### 3.2. Differential Impact of Hydrogen Peroxide Pulse on Mitochondrial and Nuclear DNA Integrity

To identify a large number of mtDNA species, we applied long-read PacBio sequencing to in vitro single-cutter-linearized DNA derived from isolated mitochondria. Although mtDNA was highly enriched in these samples, they always contained traces of nuclear DNA (3–20%). This enabled us to compare the effects of H_2_O_2_ treatment on the size distribution of DNA fragments of mitochondrial and nuclear origin (Figure 3, left and middle panels). The relative frequency of full-length mtDNA decreased dramatically 30 min after the H_2_O_2_ pulse (ratio of 0.52 in comparison to 0 h samples, *p* = 0.03), accompanied by higher relative frequencies of fragments in the range of 3–7 kb (ratio of 1.23, *p* = 0.006 for wild-type sample; ratio of 2.36, *p* = 0.01 for POLGexo^−/−^; Figure 3A,D). The fragment size distribution returned almost to the untreated condition in wild-type cells (ratio of 0.96, *p* = 0.05 for short fragments; Figure 3A) 24 h after treatment, and also partially recovered in POLGexo^−/−^ cells (ratio of 1.37 for short fragments, *p* < 0.001; Figure 3D). No such transient decrease in average fragment size after H_2_O_2_ was observed in nuclear DNA fragments (Figure 3B,E).

Treatment of the DNA samples with S1 nuclease additionally converts SSBs to DSBs. Accordingly, we observed a further increase in DNA fragmentation in both mtDNA and nuclear DNA (Figure 3G,H,J,K). However, again, no elevated frequencies of short nuclear DNA fragments were observed 30 min after H_2_O_2_ treatment (Figure 3H,K).

Surprisingly, the relative frequencies of the full-length mtDNA seemed to be higher in POLGexo^−/−^ cells (Figure 3D) than in wild-type cells (Figure 3A), which clearly contradicts our observations obtained by Southern blotting and qPCR (Figure 1 and Figure 2). An inspection of the size distributions of nuclear DNA fragments revealed that the detection probability of DNA fragments always decreased for larger sizes, and this size bias was different from reaction to reaction (Figure 3B vs. 3E). Therefore, we calculated correction factors for each size bin based on actually observed nuclear DNA sizes in each sample as compared to the sample with the lowest size bias (POLGexo^−/−^, 24 h; Figure 3E,K). These correction factors were then applied to the corresponding mtDNA fragments (Figure 3, right panels). Normalized size distributions showed that over 60% of mtDNA was intact in wild-type cells before H_2_O_2_ treatment and 24 h after treatment (Figure 3C). However, the frequency of full-length mtDNA was initially lower in POLGexo^−/−^ cells and the recovery 24 h after H_2_O_2_ treatment was only partial (Figure 3F).

### 3.3. Increased Amount of DSBs but Not SSBs in POLGexo^-/-^ Cells

Quantification of all deep sequencing-detected ends normalized to average coverage in each sample showed that approximately 2–5% of mtDNA molecules carried DSBs before H_2_O_2_ treatment in wild-type cells (Figure 4A,B), while the number of SSBs was a magnitude of order higher (Figure 4C,D). In accordance with previous reports [22,23], POLGexo^−/−^ cells already contained a large number of DSBs before H_2_O_2_ treatment (Figure 4A,B).

The H_2_O_2_ pulse led to an approximately 5-fold elevation of DSB frequency in wild-type cells, which was reversed to baseline levels after 24 h. The elevated DSB levels in the POLGexo^−/−^ cell line showed inconsistent changes upon H_2_O_2_ treatment in the two deep sequencing approaches (Figure 4A,B). The frequency of SSBs was elevated 30 min after H_2_O_2_ treatment which was reversed after 24 h, although this was not significant due to large variation and the limited number of analyzed samples. No substantial differences in SSBs were observed between wild-type and POLGexo^−/−^ cells.

### 3.4. Degradation-Specific Sequence Motifs Are Detectable in POLGexo^-/-^ Cells and upon H_2_O_2_ Damage

In addition to quantification, we aimed to see whether H_2_O_2_-induced mtDNA ends were associated with specific sequence motifs. Therefore, we determined the relative frequencies of nucleotides at positions relative to the starting or end position of detected mtDNA fragments. Ends were found to be preferentially located proximal to short GC-stretches 30 min after the H_2_O_2_ pulse, and this pattern was absent before or 24 h after the treatment in wild-type cells (Figure 5). Illumina and PacBio data were consistent. However, in POLGexo^−/−^ cells, mtDNA ends showed a pronounced association with GC-stretches at each time point (Figure 6). These observations show a striking similarity to our previously reported findings obtained from cells in which a DSB was introduced at a single site of the mtDNA by inducibly expressing a mitochondria-targeted restriction endonuclease [12]. Under those conditions, an association of ends with GC-stretches was present in wild-type cells only after 6 h of endonuclease expression but not before or 24 h after induction, while ends in POLGexo^−/−^ cells showed persistent association with GC-stretches. Combining the current results with our previous study, the GC-stretches at ends that appear upon H_2_O_2_ damage in wild-type HEK 293 cells and that are always present in POLGexo^−/−^ cells are indicative of ongoing linear mtDNA degradation, rather than the original site of DSBs.

### 3.5. De Novo mtDNA Synthesis Does Not Play a Crucial Role in Short-Term Recovery after H_2_O_2_ Damage

To determine whether increased mtDNA replication from intact mtDNA molecules contributes to the recovery observed in the wild-type cell line after the H_2_O_2_ pulse, BrdU incorporation into mtDNA was investigated. In these experiments, 20 µM aphidicolin was added to allow the halting of nuclear DNA replication [18]. Prominent bands appearing during BrdU incorporation were verified as different conformations of the mtDNA by subsequent hybridization with a mitochondria-specific probe (Appendix A). In both wild-type and POLGexo^−/−^ cell lines, BrdU was gradually incorporated into all conformations of mtDNA when applied to the cell growth medium in the absence of H_2_O_2_, although this incorporation was slightly reduced in the POLGexo^−/−^ cells (Figure 7A,B). The incorporation of BrdU into the mtDNA is consistent with ongoing cell division and accompanying replication of mtDNA to maintain constant mtDNA copy numbers. When BrdU is applied to cells along with H_2_O_2_, the BrdU incorporation was apparently lower at early time points in both wild-type and POLGexo^−/−^ cells and only became comparable to cells without H_2_O_2_ treatment after 24 h (Figure 7C,D). These data suggest that the fast recovery of intact mtDNA after H_2_O_2_ damage observed in the short time frame of 2–6 h does not depend on accelerated de novo mtDNA replication. Conversely, mtDNA replication appears transiently delayed, probably due to the lower amounts of available intact mtDNA templates. Therefore, the fast recovery of supercoiled mtDNA observed after the H_2_O_2_ pulse in wild-type cells is mainly due to the repair of SSBs.

## 4. Discussion

Mitochondrial DNA, packed into dense protein-DNA complexes (nucleoids), is associated with the matrix side of the inner mitochondrial membrane in a highly oxidative environment. The combination of abundant respiratory chain-driven superoxide generation [24,25] and the high amounts of free iron within the mitochondrial matrix [26] has a dangerous potential to generate the extremely reactive hydroxyl radical (^•^OH). Normally, the superoxide anion is rapidly converted to H_2_O_2_ by superoxide dismutases (SOD2 in the mitochondrial matrix) as part of the defense against oxidative stress. H_2_O_2_ is then either further reduced to water by the thioredoxin-2 antioxidant system (in brain mitochondria, [24]) or glutathione peroxidase and catalase (in liver mitochondria, [25,27]) but eventually serves as a potential source for the harmful hydroxyl radical produced in a reaction with free Fe^2+^ (Fenton reaction) [28,29]. The low reactivity of H_2_O_2_ makes it the only ROS that can travel larger distances within the cell. In fact, H_2_O_2_ can not only leave mitochondria, but cytosolic H_2_O_2_ is able to enter mitochondria via an aquaporin-dependent mechanism [30]. Thus, independent of the site of generation, H_2_O_2_ will likely induce oxidative damage on biomolecules in cellular compartments with high iron content, particularly within mitochondria that are important hubs of cellular iron homeostasis since iron is required for the synthesis of heme and iron-sulfur clusters [2]. In this context, it is important to mention that the intrinsic mitochondrial H_2_O_2_ splitting activity is not sufficient to cope with the maximal production rates and is highly tissue-specific [25,27].

External hydrogen peroxide has long been used as a model of oxidative damage in cultured cells [31], although its specificity was questioned, and alternative methods of oxidative damage have been investigated [32]. Here we show that, in the presence of pyruvate in the medium, the treatment with 1 mM H_2_O_2_ is well tolerated by HEK 293 cells, likely due to the fact that H_2_O_2_ readily degrades in the cell growth medium even in the absence of cells and this decay is accelerated when cells are present. Thus, the addition of H_2_O_2_ to the medium is transient and leads to a 10-min pulse of oxidative challenge. The concentrations of H_2_O_2_ that we applied to HEK 293 cells in this study are higher than the ~150 µM that was reported in vivo under the condition of ischemia-reperfusion due to intrinsic respiratory chain-driven superoxide generation [33]. Nevertheless, substantially higher local concentrations can be expected after the activation of plasma membrane NADPH oxidases of microglia or invading immune cells [34,35].

In our model system of HEK 293 cells, hydrogen peroxide induces accumulation of both DSBs and SSBs in mtDNA, which reaches a maximum of half an hour after the H_2_O_2_ pulse. In contrast, H_2_O_2_-induced fragmentation of the nuclear genome was not detectable, as demonstrated by the unaltered size distribution of long single-molecule-derived nuclear sequences detected by the PacBio long-read sequencing technique. This is consistent with the above-mentioned assumption that H_2_O_2_ can reach various cellular compartments but leads to oxidative damage, primarily in those with high free iron concentrations. Hydroxyl radicals locally generated by the iron-dependent Fenton reaction can attack the sugar backbone of mtDNA [10], leading to SSBs, and likely also to DSBs through a second hit or through replication stalling at SSBs or modified bases. The hydroxyl radical can also oxidize single nucleotides; however, we did not address this question in this study. It has been shown that mitochondrial BER is able to remove oxidized nucleotides from the mtDNA [11]. The excision of the oxidized base and the subsequent removal of the remaining sugar moiety generates a gap, i.e., an SSB, which is then re-sealed by the concerted action of the polymerase activity of POLG, various 5′-3′ exonucleases and, finally, the mitochondrial ligase LIG3. Our data show that SSB repair as part of the BER pathway after oxidative damage is functional both in wild-type and POLGexo^−/−^ cells and that the loss of the 3′-5′ exonuclease activity of POLG does not interfere with the SSB repair. Using our technique to assess SSBs, we cannot distinguish between SSBs on circular and linear mtDNA species. This distinction is hardly relevant in wild-type cells in which the vast majority of mtDNA exists in the form of supercoiled circular DNA. In opposite to this, exonuclease-deficient POLG is known to lead to the accumulation of linear mtDNA species [22,23]. When SSBs on such linear mtDNA molecules are locally repaired, this repair does not generate fully intact supercoiled mtDNA but reduces the number of SSBs on both circular and linear mtDNA.

Wild-type HEK 293 cells not only survive the H_2_O_2_ pulse but are able to partially recover the integrity of their mitochondrial genomes within hours, as indicated by the reappearance of supercoiled mtDNA species on Southern blots. However, we did not detect any signs of boosted replication in BrdU incorporation assays, which indicates that mtDNA replication does not play a central role in the early phase of recovery from oxidative damage-induced mtDNA breaks.

Unlike the response to SSBs, there was a dramatic difference in the dynamics of generation and clearance of DSBs between wild-type and POLGexo^−/−^ cells. This is not surprising given the fact that we and others demonstrated that the 3′-5′ exonuclease activity of POLG is a central part of the machinery that is responsible for the rapid degradation of linear mtDNA species [12,36,37]. In addition, another hallmark of ongoing mtDNA degradation, a preferential localization of mtDNA ends proximal to GC-stretches, was detected 30 min after oxidative damage. We previously identified such sequence motifs as transient pausing sites of degradation of linear mtDNA [12]. This pattern was absent 24 h after the H_2_O_2_ pulse in wild-type cells, indicating successful completion of the removal of fragmented mtDNA.

Interestingly, POLGexo^−/−^ cells showed similar sequence patterns of mtDNA ends prior to H_2_O_2_ damage as well as at 30 min or 24 h after the H_2_O_2_ pulse. This apparently contradicts our previous finding that the loss of the 3′-5′exonuclease activity of POLG inhibits linear mtDNA degradation [12]. In our previous model of DSB formation, a mitochondria-targeted restriction endonuclease was used to cleave mtDNA at a single position. Although gross degradation was inhibited, both by inactivation of the 3′-5′ exonuclease activity of POLG or knocking out the mitochondrial 5′-3′ exonuclease MGME1, an inspection of the vicinity of the cutting site reveals that, unlike the MGME1 knock-out, exonuclease deficient p.D274A POLG still exhibits a small residual activity of linear mtDNA degradation (cf. Figure 3a in [12]). Of note, the vast majority of linear mtDNA species in POLGexo^−/−^ cells exist independent of the H_2_O_2_ pulse, and the observed ends represent intermediates of severely impaired degradation of continuously generated linear mtDNA species.

In contrast to previous approaches, which utilized mitochondria-targeted endonucleases to artificially generate double-strand breaks in the mtDNA [12,36], in the present work, we used a pulse with the natural oxidant hydrogen peroxide to induce the generation of mtDNA single-strand and double-strand breaks. After this oxidative insult, the integrity of mtDNA was rapidly restored by fast degradation of linear mtDNA and rapid repair of single-strand breaks, while mtDNA re-synthesis by replication was identified as a much slower process (Figure 1).

Rapid degradation of linear mtDNA is, therefore, an important quality control pathway of mtDNA because it precludes the potential formation of harmful mtDNA deletions from accumulating linear mtDNA molecules by non-homologous end-joining [36]. However, under conditions when the degradation of linear mtDNA is not efficient enough to completely remove all linear mtDNA fragments, somatic mtDNA deletion generation could result (cf. Figure 1). That could explain findings of low-level mtDNA deletions in pathological brain tissue affected by inflammation [38,39] since invading immune cells or activated microglia have been identified as potential sources of hydrogen peroxide [34,35].

## 5. Conclusions

Our data highlight the interplay and timing of repair, degradation, and re-synthesis of mtDNA after transient oxidative damage. Furthermore, the results demonstrate the essential role of the rapid linear mtDNA degradation pathway for mtDNA quality control to avoid somatic mtDNA deletion formation.

## Data Availability

The data presented in this study are available in the article and Appendix A.

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
