# Peer review of "The Fate of Oxidative Strand Breaks in Mitochondrial DNA"

_antioxidants, 2023, doi:10.3390/antiox12051087_

Round 1
Reviewer 1 Report
In this article by Trombly et al., the authors described the fate of mitochondrial DNA in response to oxidative damage induced by a transient pulse of hydrogen peroxide in human HEK 293 cells, wild-type or deficient for exonuclease POLG. By using different complementary approaches ( southern-blotting, q-PCR, short-read and long-read sequencing), the authors found that a transient exposure to oxidative stress induces accumulation of mitochondrial single-strand and double-strand breaks mitochondrial DNA fragments that generate linear mitochondrial DNA fragments with GC-stretches at the ends. The linearized mitochondrial DNA is rapidly degraded and intact supercoiled mitochondrial DNA species are recovered through the repair of single-strand break rather than de-novo mitochondrial DNA synthesis. They found that linear mitochondrial DNA fragments persist in the endonuclease-deficient POLG mutant upon hydrogen peroxide pulse while the repair of single-strand breaks was not affected. Altogether these data indicate that upon oxidative damage, oxidative strand breaks are rapidly processed with fast repair of single strand breaks and degradation of double-strand breaks-generated linear mitochondrial DNA to prevent formation of somatic mitochondrial DNA deletions.
To conclude, I think this paper is full in line with the topics of Antioxidants and this study should be of interest to a broad audience including specialists of mitochondrial DNA and DNA repair fields. However, improvement of the robustness of some of their analyses (see above, lack of statistical significances or, in some cases, no biological replicates to support their conclusions (i.e. Figure S4, Figure 3, Figure 4, Figure 7) and some points listed below are needed to be addressed before publication.
Specific comments:
Line 32: In the introduction, I think references to previous papers are lacking such as paper from Shokolenko et al. 2009 (PMID PMID: 19264794 ) showing that oxidative stress (i.e. H2O2 treatment) induces degradation of mitochondrial DNA. Could the authors introduce more in details previous works about impact of oxidative damage on mitochondrial DNA.
Line 189, Figure S4: The figure S4 show no indication of statistical significance between the different quantifications. Did the authors perform statistical tests for the southern blot quantifications? Could the authors add the statistical marks on the graphs. Without statistical tests, it is not really robust to draw conclusions such as lines 194-200 and 201-216.
Line 217, Figure 1: Could the authors combine representative southern blot images with the corresponding quantification histograms on the main figure.
Line 242, Figure 2: I think a drawing of the primers positions on mitochondrial genome could be helpful.
Line 263: “we observed a further increase…. (Figures 3G, H, J and K)”. There is no marks of statistical significance.
Line 279, Figure 3: This figure is not very readable; Could the authors enlarge this figure,? (for example, the legend “Oh, 0.5h and 24h” could be moved at the bottom and the title of the Y axis could be vertically rotated to enlarge the histograms) . There is no indication of the number of biological replicate for this experiment and no statistical significances indicated. Did the author perform statistical tests for this experiment?
Line 297, Figure 4, panels B, C and D: There is no biological replicate presents for the experiment performed on POLGexo-/- cells (n=1). The authors conclude line 312 (and line 305) that no substantial differences for single strand breaks formation between Wild-type and POLGexo-/- cells is observed. Thus, to my opinion, it is not possible to give a conclusion on this experiment with only on replicate and without performing a statistical test. If the test give no statistical differences for C and D graphs, it will be better to indicate “ns” for non-statistically significant p value, in this case.
Line 332, Figure 5: Could the authors enlarge this figure ?
Line 341, Figure 6: Could the authors enlarge this figure ?
Figure 7: This figure lacks indications of statistical significance. Did the authors perform statistical tests to establish their conclusions ? Could the authors add the significance marks on the graphs.
Line 392: For the discussion part, It would have been helpful to have a summary of the main results of the study at the beginning of the discussion. To my opinion , the first paragraph (lines 393-411) is not really a discussion about the results of the study. Maybe the place of this paragraph would have been better in the introduction, or the author should better show the connections with their results.
Minor points
Line 16: I suggest to change “a transient H2O2 pulse” by “a transient hydrogen peroxide (H2O2 pulse)”
Line 18: I suggest to add a comma after “pulse”
Line 34: : I suggest to change “mutation” by “mutations
Line 59 and/ or line 65: could the author define in the text what are the HEK293 cells (i.e. immortalized human embryonic kidney)
Lines 112, 138, 156: could the author provide a reference for these methods?
Line 188: could you rephrase this sentence, it is a bit confusing “We performed Southern blotting of agarose electrophoresis separated whole cell DNA under non-denaturing conditions…”?
Line 235-236: A word is missing: I suggest to add ‘is” after “The larger the observed Ct difference between both amplicons»
Line 275: I suggest to remove “s” from “sizes”
Line 309: “inconsistant” should be corrected by “inconsistent”
Line 321: “consistant” should be corrected by “consistent”
Line 328: “persistant” should be corrected by “persistent”
Line 373: A word is missing: I suggest to add “to” before “ visualize BrdU…”
Author Response
Specific comments:
Line 32: In the introduction, I think references to previous papers are lacking such as paper from Shokolenko et al. 2009 (PMID PMID: 19264794 ) showing that oxidative stress (i.e. H2O2 treatment) induces degradation of mitochondrial DNA. Could the authors introduce more in details previous works about impact of oxidative damage on mitochondrial DNA.
: Done, references added (lines 56-59).
Line 189, Figure S4: The figure S4 show no indication of statistical significance between the different quantifications. Did the authors perform statistical tests for the southern blot quantifications? Could the authors add the statistical marks on the graphs. Without statistical tests, it is not really robust to draw conclusions such as lines 194-200 and 201-216.
: Done, Fig. S4 with statistics has been combined with Fig. 1
Line 217, Figure 1: Could the authors combine representative southern blot images with the corresponding quantification histograms on the main figure.
: Fig. 1 has been replaced by combined graph.
Line 242, Figure 2: I think a drawing of the primers positions on mitochondrial genome could be helpful.
: Fig. 2 has been modified accordingly.
Line 263: “we observed a further increase…. (Figures 3G, H, J and K)”. There is no marks of statistical significance.
: Results of the statistical analysis has been included in the main text (lines 287-293).
Line 279, Figure 3: This figure is not very readable; Could the authors enlarge this figure,? (for example, the legend “Oh, 0.5h and 24h” could be moved at the bottom and the title of the Y axis could be vertically rotated to enlarge the histograms) . There is no indication of the number of biological replicate for this experiment and no statistical significances indicated. Did the author perform statistical tests for this experiment?
: Fig. 3 has been enlarged, the number of biological replicates has been indicated in the legend.
Line 297, Figure 4, panels B, C and D: There is no biological replicate presents for the experiment performed on POLGexo-/- cells (n=1). The authors conclude line 312 (and line 305) that no substantial differences for single strand breaks formation between Wild-type and POLGexo-/- cells is observed. Thus, to my opinion, it is not possible to give a conclusion on this experiment with only on replicate and without performing a statistical test. If the test give no statistical differences for C and D graphs, it will be better to indicate “ns” for non-statistically significant p value, in this case.
: We have included the ns differences in the figure. Regarding SSBs, additional experimental data with the qPCR approach presented in Fig. 2 for S1 nuclease-treated samples have been included in the text (lines …). If comparing the 0 hour and 24 hour samples no difference in the SSBs between wt and POLGexo-/- cells has been observed.
Line 332, Figure 5: Could the authors enlarge this figure ?
: Done.
Line 341, Figure 6: Could the authors enlarge this figure ?
: Done.
Figure 7: This figure lacks indications of statistical significance. Did the authors perform statistical tests to establish their conclusions ? Could the authors add the significance marks on the graphs.
: We performed statistical testing but not achieved statistical significance. Therefore, we reformulated the text accordingly.
Line 392: For the discussion part, It would have been helpful to have a summary of the main results of the study at the beginning of the discussion. To my opinion , the first paragraph (lines 393-411) is not really a discussion about the results of the study. Maybe the place of this paragraph would have been better in the introduction, or the author should better show the connections with their results.
: The main results of the study are summarized in the second and third paragraph of discussion. The first paragraph justifies the use of H2O2 as a ‘physiological’ oxidant and is in our opinion closely related to the second paragraph.
Minor points
Line 16: I suggest to change “a transient H2O2 pulse” by “a transient hydrogen peroxide (H2O2 pulse)”
: Changed accordingly.
Line 18: I suggest to add a comma after “pulse”
: Changed accordingly.
Line 34: : I suggest to change “mutation” by “mutations
: Changed accordingly.
Line 59 and/ or line 65: could the author define in the text what are the HEK293 cells (i.e. immortalized human embryonic kidney)
: Changed accordingly.
Lines 112, 138, 156: could the author provide a reference for these methods?
: The missing references have been included.
Line 188: could you rephrase this sentence, it is a bit confusing “We performed Southern blotting of agarose electrophoresis separated whole cell DNA under non-denaturing conditions…”?
: We rephrased the sentence.
Line 235-236: A word is missing: I suggest to add ‘is” after “The larger the observed Ct difference between both amplicons»
: Done.
Line 275: I suggest to remove “s” from “sizes”
Done.
Line 309: “inconsistant” should be corrected by “inconsistent”
: Done.
Line 321: “consistant” should be corrected by “consistent”
: Done.
Line 328: “persistant” should be corrected by “persistent”
: Done.
Line 373: A word is missing: I suggest to add “to” before “ visualize BrdU…”
: Done.
Reviewer 2 Report
In this manuscript, the authors demonstrate that H2O2 induced oxidative damage to mitochondrial DNA (mtDNA) results in linear fragmentation of mtDNA that is corrected over time, likely through rapid repair of single strand breaks and degradation of linear fragments resulting from double strand breaks.
The study is well executed and described, and the writing and presentation is clear and precise. My only possible suggestion is that more care could be taken with the qPCR results; for example, determination of the efficiencies of primers used and taking these efficiencies into account would result in more quantitatively reliable qPCR results.
Author Response
The study is well executed and described, and the writing and presentation is clear and precise. My only possible suggestion is that more care could be taken with the qPCR results; for example, determination of the efficiencies of primers used and taking these efficiencies into account would result in more quantitatively reliable qPCR results.
: The efficiencies of PCR primers used for qPCR have been provided at the end of paragraph 2.6.
Reviewer 3 Report
In the paper entitled “The fate of oxidative strand breaks in mitochondrial DNA” the authors evaluate the effects of a pulse of H2O2 on mitochondrial DNA in wt HEK239 and in a POLG mutant deficient in 3’-5’ exonuclease activity, previously constructed. The authors studied the dynamics of DNA damages induced by H2O2 and the interplay between repair and DNA replication. They observed that in WT cells SSB induced by H2O2 are rapidly repaired and DSB formed fragments degraded. Meanwhile they demonstrated that mtDNA replication is not involved in the repair of H2O2-induced mtDNA damages, but it is slower compared to WT cells probably to prevent duplication of mutated DNA. H2O2 treatment in exonuclease-deficient POLG p.D274A mutant cells results in persistence of linear mtDNA fragments without affecting the repair of SSBs.
Overall the paper is well written, experiments are well mapped out and results are convincing, however, the manuscript contains some minor concerns that need to be addressed before publication.
Minor concerns
Figure 2. in the figure are reported the asterisks of significance, but it is not explained if data are compared with control (0) or with WT
Line 264: there is the description of experiments involving S1 nucleases. To improve the comprehension of the experiment I would suggest that the authors add a little explanation of the activity of S1 nuclease. At a first reading is not clear the information gained by this experiment.
Line 441-443: It should be better explained why BER is considered functional in POLG p.D274A mutant cells. If damaged mtDNA persists in the cell and there is not recovery of supercoiled DNA as for WT cells, how can we say that BER is functional?
Author Response
Minor concerns
Figure 2. in the figure are reported the asterisks of significance, but it is not explained if data are compared with control (0) or with WT
: In the revised Fig. 2 the significance comparison (with wild-type) is clarified by a line.
Line 264: there is the description of experiments involving S1 nucleases. To improve the comprehension of the experiment I would suggest that the authors add a little explanation of the activity of S1 nuclease. At a first reading is not clear the information gained by this experiment.
: An explanation for S1 nuclease action has been provided in this paragraph.
Line 441-443: It should be better explained why BER is considered functional in POLG p.D274A mutant cells. If damaged mtDNA persists in the cell and there is not recovery of supercoiled DNA as for WT cells, how can we say that BER is functional?
: To the paragraph 3.1 (at the end) we included data from additional experiments from S1 nuclease-treated samples showing no difference between wild-type and the POLG p.D274A mutant cells at the 0 hour and the 24 h time point. From these experiments a functional BER in the POLGexo-/- mutant can be concluded. To explain the missing recovery of supercoiled mtDNA in POLG we added a small paragraph in the discussion section (lines 480-486).
Round 2
